# Caregivers’ Expectations on Possible Functional Changes following Disease-Modifying Treatment in Type II and III Spinal Muscular Atrophy: A Comparative Study

**DOI:** 10.3390/jcm12134183

**Published:** 2023-06-21

**Authors:** Maria Carmela Pera, Giorgia Coratti, Jacopo Casiraghi, Chiara Bravetti, Alessandro Fedeli, Milija Strika, Emilio Albamonte, Laura Antonaci, Diletta Rossi, Marika Pane, Valeria Ada Sansone, Eugenio Mercuri

**Affiliations:** 1Department of Life Science and Public Health, Pediatric Neurology, Università Cattolica del Sacro Cuore, 00168 Rome, Italy; 2The NEMO Center in Rome, Fondazione Policlinico Universitario Agostino Gemelli IRCCS, 00168 Rome, Italy; 3The NEMO Center in Milan, Neurorehabilitation Unit, University of Milan, ASST Niguarda Hospital, 20162 Milan, Italy

**Keywords:** spinal muscular atrophy, quality of life, caregiver, clinical trials

## Abstract

*Background:* The primary aim of this study was to explore current caregivers’ expectations on possible functional changes following treatment in comparison to data obtained in the pre-pharmacological era. *Methods:* A questionnaire, previously used in 2016, was administered to caregivers of type II and III SMA patients of age between 3 and 71 years, and to patients over the age of 13 years. The questionnaire focuses on (1) caregivers and patients expectations, (2) meaningfulness of the changes observed on the functional motor scales, and (3) their willingness to be enrolled in a clinical trial. A comparative study was performed with data obtained using the same questionnaire soon before the advent of disease-modifying therapies. *Results*: We administered the questionnaire to 150 caregivers. When comparing current caregiver data to those obtained in 2016, the most obvious differences were related to disease perception over the last year (stability: 16.5% in 2016 vs. 43.6% in 2022; deterioration 70.5% vs. 12.8%, and improvement: 12.9% vs. 43.6%) and expectations from clinical trials with higher expectations in 2022 compared to 2016 (*p* < 0.001). Forty-five of the 150 in the current study were caregivers of patients above the age of 13. In these 45 the questionnaire was also administered to the patient. No difference was found in responses between patients and their caregivers. *Conclusions*: Both carers and patients reported that even small changes on functional scales, similar to those reported by clinical studies and real-world data, are perceived as meaningful. Comparing the recent responses to those obtained in 2016, before pharmacological treatment was available, we found significant changes in caregivers’ perception with increased expectations. These findings will provide a better understanding of the patients’ expectations and facilitate discussion with regulators.

## 1. Introduction

Spinal muscular atrophy (SMA) is an autosomal-recessive disorder characterized by motor neuron loss in the spinal cord, leading to progressive proximal muscle weakness and atrophy [1]. SMA is due to biallelic mutations in the survival motor neuron 1 (*SMN1*) gene, located on chromosome 5q13 [2]. Historically, clinical phenotypes have been subdivided into four subtypes with different clinical severities on the basis of age of onset and maximum motor function acquired. Type I is associated with profound weakness occurring in the first 6 months of life and inability to sit. Type II SMA has an intermediate phenotype, with disease onset between 6 and 18 months of age. Patients are able to maintain the sitting position but never acquire the ability to walk and, by the age of puberty, they have lost most of their functional abilities. In type III the onset occurs after 18 months of age and although patients reach the ability to walk, this is generally associated with some functional decline, and loss of the ability to walk in a significant number of type III patients. Type IV is a rare (<1% of SMA cases), adult form with the mildest phenotype in which onset is after the age of 18 years [3,4,5]. In the last few years, the advent of disease-modifying therapies (DMT) has dramatically changed the disease course of SMA, with an increasing number of patients showing improvements or at least stabilization of motor and respiratory functions. The changes in progression have led to unexpected results, such as sitting in type I patients or walking in type II so the lines between these categories are now significantly blurred [6,7].

The question has arisen whether the new changes have also produced a change in patients’ and caregivers’ perspectives [8,9]. This observation appears to be relevant as this approach, in combination with the objective clinician-administered outcome measures, has been strongly encouraged by the United States Food and Drug Administration (FDA) and by the European Medicine Agency (EMA) [10,11,12].

In 2016, soon before the advent of the new therapies, as part of an international effort we used a structured questionnaire to explore patients’ and caregivers’ perspectives and expectations on possible functional changes in relation to current or future clinical trials [13].

The primary aim of this study was to use the same questionnaire to compare the current responses to those obtained in our previous study at the time DMTs were not available.

## 2. Materials and Methods

This study has been conducted in two Italian tertiary care centers: IRCCS Policlinico Gemelli in Rome and Centro Clinico Nemo in Milan. The study was approved by the Institutional Review Board (Ethics Committee) in each center. Written informed consent was obtained from all participants (or guardians of participants) in the study. Study participants did not receive any form of compensation.

The study included all patients diagnosed with SMA5q with clinical phenotypes consistent with type II and III SMA according to the original SMA classification [3,4]. We enrolled consecutive patients attending our centers between January and June 2022 who agreed to participate in the study by signing voluntary informed consent. In the case of minors, both parental or guardian consent and the assent of the minors were obtained. The study was proposed to individuals of all age groups, including children and adults, irrespective of age, functional or pharmacological treatment status. 

Structured questionnaires were administered by trained clinicians using the same tool and the same procedures used in 2016 [13] with an additional two questions. The interviews lasted 20–45 min on average.

For patients ≥13 years of age, in addition to their carers’ responses, the questionnaire was also independently filled by the patients. The choice of 13 years is in Italy considered the age when teenagers have the ability to provide meaningful answers. Many psychometric studies also use the age of 13 to collect data on minor patients as from this age there is more awareness of the disease, making the responses more reliable than those obtained in younger children [14]. 

Since the questions were tailored according to the motor functional level of the individual patient, the questionnaires were performed within 3 months of a recent clinical functional assessment. 

In the first part of the questionnaire, caregivers and patients were asked about the trend of motor skills in the previous year and their expectations for the next two years (see Appendix A for details of the questionnaire). In the second part, questions were related to the results of the functional scales (HFMSE and RULM). In both HFMSE and RULM the items follow a hierarchical order with increasing difficulty, built on the frequency distribution of findings observed in a large cohort of SMA patients [15,16,17,18,19]. On the basis of the first three items not acquired on the two scales (score 0) or partially acquired (score 1), we asked patients/caregivers if they would consider the possibility to participate in a clinical trial with the prospect of achieving/improving one or more of these items. 

In the third part, the caregivers were asked whether they would consider having their child take part in a potential trial in the presence of mild side effects (defined as headache, abdominal discomfort, and tachycardia). 

In order to better understand patients’ and caregivers’ attitudes toward new treatments outside clinical trials, we added two questions that were not present in the original questionnaire, as therapies were not available in 2016. The new questions investigate if and how the perception of the disease has changed with the advent of new therapies and what the expectations regarding the new therapies are.

### Statistical Analysis

The cohort was subdivided into caregivers and patients and subgrouped by type, motor functional status, and participation in the previous study [13]. 

To determine the sample size for the study, data was derived from the Italian ISMAR registry [20] and included 245 patients diagnosed with SMA II and III from the two centers as of December 2021. In order to achieve a 95% confidence interval with an alpha value of 0.05, a sample of 150 individuals was deemed sufficient to effectively survey the population.

Descriptive statistic (N, mean, SD, range) was used to summarize data on age, gender, SMA type, and motor functional status. For the purpose of this study, adult patients were identified as individuals who were over 18 years old at the time of providing their consent. The motor functional status of the patients was classified as follows: “sitters” were defined as individuals capable of sitting independently for a minimum of 3 s, while “ambulant” referred to individuals who could walk independently for a distance of at least 10 m.

The first part of the analysis was dedicated to the results of the 2022 questionnaire, including a large cohort of patients and carers. A chi-square test/Fisher’s exact test of independence was used to compare the cohorts’ responses distribution between age (pediatric vs. adult), participant’s status (caregivers, patients), and motor functional status (ambulant, non-ambulant). As per statistical procedure, we used the Chi-square test if all of the expected cells were greater than 5, while we used Fisher’s exact test of independence if an expected number was less than 5. The *p*-value was set at <0.05 for all the analyses.

As part of the study’s primary aim, we also explored possible differences between the cohorts’ responses obtained in the current study and those recorded in the 2016 study using the same questionnaire [13]. As in the 2016 study, only caregivers of pediatric patients (0–17 years) were included, this analysis was performed using similar inclusion criteria in the current cohort. 

Regarding the study’s characteristics, no missing data were observed, and consequently, no imputation was required.

## 3. Results

Table 1 shows details of the SMA patients included in this study. All received treatment with a disease-modifying therapy (DMT), with a mean duration of 3.13 years (SD: 0.75). This is different from the previous study in which none of the patients were treated with DMTs or were enrolled in clinical trials.


Caregivers


Of the 150 caregivers, 131 were parents and 19 were siblings or partners. 

Forty-seven of the 150 caregivers had a degree, 68 had a secondary and 35 had a primary education degree. Only 5 caregivers were in the healthcare field. 

All 150 of the caregivers in the new cohort answered all the questions. Details are reported below. 

-*Speaking of motor skills, please indicate if, in the last year, your child remained stable, deteriorated, or improved*: 50% of the caregivers reported stability, 16.67% deterioration, and 33.33% improvement.-*What do you expect in the next two years?* 51.33% of the caregivers anticipated a stable course, 6.67% a deterioration, and 42% an improvement.-*Would you agree to have your child participate in a potential clinical trial if, in the absence of side effects or with the possibility of minimal side effects, the prospect was to slow down or to improve or stop the deterioration of motor function?* 68.67% of the caregivers would participate if the treatment slowed down deterioration, 77.33% if it would stop deterioration, and 96% if the treatment would produce an improvement.-*Would you consider having your child participate in a clinical trial if it was offered the prospect of achieving at least 1, 2, or more than 2 of the following activities on the HFMSE/RULM?* At the time of this question, the first three items that had not been fully achieved on the scale were shown to the caregivers as the activities that could be possibly achieved. Approximately 60% would participate in a clinical trial if at least 1 or 2 of the 3 activities could be achieved, with 99% agreeing to participate if more than 2 activities could be achieved.

Table 2 reports the results of the analysis subdivided by the scale and starting point in the new cohort assessed in 2022.


*New questions:*


-*With the advent of new pharmacological therapies, how has your perception of your child’s disease changed?:* 22% of the caregivers reported no change, 74% felt more positive, 0% felt more negative, and 4% was uncertain.-*What are your expectations regarding the new pharmacological therapies?:* 16% reported no change, 78.77% were more positive, 2% were more negative and 3.33% were uncertain.


Caregivers subdivided by patient’s age


We analyzed the new caregiver cohort subdivided by whether the patients were below (n = 78) or above (n = 72) the age of 18 years. 

Significant differences were found in two of the 10 questions:-Clinical course perception over the previous year: (x^2^ (2,N = 150)= 7.906, *p* = 0.019), with the carers of pediatric patients reporting more cases of improvement than the carers of adult patients.-Perception of the disease has changed with the advent of new therapies: (x^2^ (2, N = 150) = 7.357, *p* = 0.025), with carers of pediatric patients having a more positive approach than the carers of adult patients.
Comparison between patients’ and caregivers’ responses

Forty-five consecutive patients ≥13 years of age also filled out the questionnaire (Table 1). For all 45 the questionnaire was also independently filled by the caregiver. 

There were no significant differences between the patients’ and caregivers’ perspectives.

Details can be found in Table 3.


*
Comparison between 2016 and 2022 caregivers’ responses
*


Seventy-eight of the 150 caregivers participating in the current study had similar inclusion criteria to the 2016 study and were used for comparison. Of the 78 caregivers participating in the current study, 58 had previously participated in the 2016 study.

The population’s characteristics in terms of age, SMA type, functional level, and gender are shown in Table 4.

Significant differences between 2016 [13] and 2022 responses were found in 6 of the 7 (86%) close-ended questions available in both questionnaires (Table 5, Figure 1A,B).

## 4. Discussion

The positive outcome of the available therapeutic approaches for type II and III SMA patients in clinical trials [20,21,22,23] and real-world data [24,25,26] has highlighted the need to better understand if this has produced a change in caregiver’s perceptions of the disease [27,28,29,30,31,32]. The need to capture patients’ and caregivers’’perspectives on the impact of the new treatments [33,34,35,36,37] is not only acknowledged by clinicians and families but has also been strongly encouraged by the United States Food and Drug Administration (FDA), by the European Medicine Agency (EMA) [10,11,12] and endorsed by the SMA community [38,39,40].

We were in a privileged position to capture possible changes following the advent of the new therapies as in 2016 we had already used a questionnaire assessing carers’ perceptions soon before the advent of the new therapies. The questionnaire used in both studies had some innovative aspects as rather than just asking general open questions, these were tailored according to each participant’s specific functional level based on their HFMSE and RULM score [17,18,19]. This approach allowed us to ask targeted individually relevant questions to patients and caregivers, realistically evaluating possible future changes in relation to their functional ability at the moment. 

We first analyzed the results of the 2022 questionnaire. A larger inclusion of adult patients allowed us to establish differences between carers of pediatric and adult patients, both in terms of clinical course perception over the previous year and in the overall perception of the disease following the advent of new therapies. Carers of younger individuals had a more positive perception and reported an overall improvement during the previous year compared to carers of adult patients [6,7]. These findings are concordant with the results from real-world studies also reporting better clinical responses in younger type II and III patients [24,25,26] compared to adults in whom more often there were reports of overall stability. 

When we performed a comparative analysis of the 2022 study with the already published 2016 results, there were significant changes in 6 of the 7 questions assessing carers’ perceptions. When asked to comment on the disease course in the previous year, in 2016 the majority of caregivers (70.5%) felt that over the last year, their child had shown deterioration, this was reported in only 12.8% in 2022. As a corollary of this, while in 2016 an improvement was reported in only 13%, in 2022 the percentage raised to 43.6%. Not surprisingly, similar findings were found when caregivers were asked about their expectations for the next 2 years.

These changes were also reflected in the reply to the question addressing participation in a clinical trial. While in 2016 there was a high of patients wishing to be enrolled even with the possibility of achieving one or two additional functional activities (66.9%/80.6%), in 2022 this percentage lowered to 59% and 61.6%, this suggesting that for carers and patients stability of minimal improvement was somehow already available and only a more marked improvement would justify their enrolment in a clinical trial. Nevertheless, from 2016 to 2022 the participation of patients appears not conditioned by the possible presence of minimal side effects of the drug, and this is shown by the only non-significant question in the comparison between the two populations.

While in 2016 we mainly used the questionnaire in caregivers, in the current study, we had the opportunity to include 45 patients (all older than 13 years) and were able to compare their responses to those obtained from their caregivers who also had independently filled the questionnaire. Interestingly, there were no significant differences between the two cohorts. These findings are at variance with previous studies in SMA patients suggesting that carers had more negative illness perceptions about SMA than their children, especially with regard to the physical symptoms and the emotional impact of the disease [41,42,43,44]. Similar findings had also been reported among parents and children with cancer, diabetes, and asthma [45,46,47] and in partner-caregivers of adult patients [48,49,50], making it less likely that immaturity and limited understanding of the disease of a child could be exclusively responsible for this discrepancy.

The better concordance in our cohort is likely to reflect the changes observed following the introduction of the new therapies that are also confirmed by the use of functional scales. One of the limitations of our study is that all our patients are followed in tertiary care centers and may therefore be not fully representative of the whole SMA population. 

These results should therefore be interpreted with caution also because they were obtained in a single country. As suggested by Fischer et al. [41], further studies merging populations from different countries and cultural backgrounds are needed to better assess this fundamental aspect of disease management.

## 5. Conclusions

The results indicate an overall more positive attitude both in terms of recent disease courses and expectations. Carers now perceive a higher chance of at least remaining stable as opposed to 2016 and although the great majority, regardless of age or functional level, would still consider participating in a clinical trial even if the prospects were limited to stability of the disease, there was a clear shift with increased expectations related to new trials. The results of the current study updates our knowledge of the impact of DMTs on patient and caregiver perspectives and may facilitate shared decision-making in clinical practice. In a historical moment of strong changes for SMA patients, it is essential to highlight the point of view of patients and caregivers with respect to the expectations towards the therapies currently used. Our findings suggest that the magnitude of changes reported in the literature was perceived as meaningful from a carer/patient perspective. Our findings will provide a better understanding of the patients’ views on this topic and facilitate discussion with regulators.

## Figures and Tables

**Figure 1 jcm-12-04183-f001:**
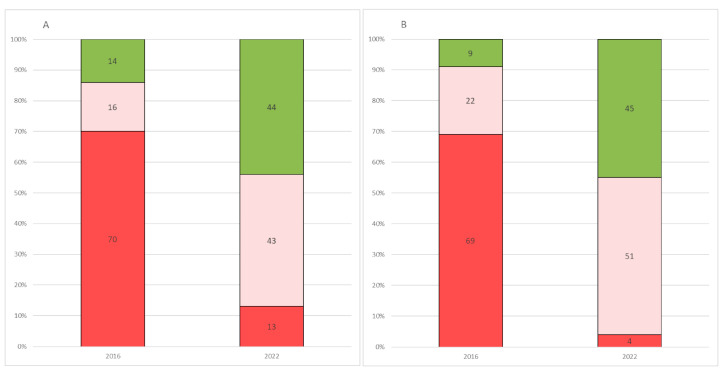
**Comparison between 2016 and 2022 on questions 1 and 2.** (**A**) Distribution of caregivers’ responses on disease perception over the previous year, recorded in 2016 and 2022. (**B**) Distribution of caregivers’ responses on their expectations for the next 2 years, recorded in 2016 and 2022. Key to figure: Red = deterioration; Light red = stability; Green= improvement.

**Table 1 jcm-12-04183-t001:** Population characteristics at the interview. Key to the table: All values refer to the person affected with SMA.

	Caregiver (n = 150)			Patients (n = 45)		
	All	Adult (n = 72 )	Pediatric (n = 78)	All	Adult (n = 33)	Pediatric (n = 12)
**Patient’s age (years) ^1^**	3–71 (20.18; 14.24)	18–71 (31.11; 13.32)	3–17 (10.08; 3.77)	14–62 (24.09; 10.22)	18–62 (27.18; 10.31)	14–17 (15.58; 0.99)
**Gender ^2^**	82 F (54.66%) 68 M (45.33%)	40 F (55.55%) 32 M (44.44%)	42 F (53.84%) 36 M (46.15%)	26 F (57.77%) 19 M (42.22%)	18 F (54.54%) 15 M (45.45%)	8 F (66.66%) 4 M (33.33%)
**Sma Type ^2^**						
**Sma Type II**	90 (60.00%)	36 (50%)	54 (69.23%)	24 (53.33%)	17 (51.51%)	7 (58.33%)
*Ambulant*	0 (0.00%)	0 (0.00%)	0 (0.00%)	0 (0.00%)	0 (0.00%)	0 (0.00%)
*Non Ambulant*	90 (100.00%)	36 (100.00%)	54 (100.00%)	24 (100.00%)	17 (100%)	7 (100.00%)
**Sma Type III**	60 (40.00%)	36 (50%)	24 (30.76%)	21 (46.66%)	16 (48.48%)	5 (41.66%)
*Ambulant*	26 (43.33%)	11 (30.55%)	15 (62.5%)	13 (61.90%)	11 (68.75%)	2 (40%)
*Non Ambulant*	34 (56.66%)	25 (69.44%)	9 (37.5%)	8 (38.09%)	5 (31.25%)	3 (60%)

All values refer to the person affected with SMA. ^1^ Range (mean; standard deviation), ^2^ N (%); F = Female, M = Male.

**Table 2 jcm-12-04183-t002:** Perception on the meaningfulness of improvements/acquisition on the HFSME or the RULM justifying the willingness to enter a clinical study if 1, 2, or more than 2 activities are improved/acquired on the basis of the first three items scoring 0/1 in the new cohort assessed in 2022.

		Whole Cohort (N; %)
**HFMSE** **Acquisition from score 0** Not applicable in 11 patients	I would not participate	1; 0.72%
	At least 1 activity	83; 59.71%
	Two activities	92; 66.19%
	More than 2 activities	138; 99.28%
**HFMSE** **Improvements from score 1** Not applicable in 35 patients	I would not participate	2; 1.74%
	At least 1 activity	70; 60.87%
	Two activities	81; 70.43%
	More than 2 activities	113; 98.26%
**RULM** **Acquisition from score 0** Not applicable in 31 patients	I would not participate	1; 0.84%
	At least 1 activity	67; 56.30%
	Two activities	77; 64.71%
	More than 2 activities	118; 99.16%
**RULM** **Improvements from score 1** Not applicable in 30 patients	I would not participate	1; 0.83%
	At least 1 activity	78; 65%
	Two activities	85; 70.83%
	More than 2 activities	119; 99.17%

**Table 3 jcm-12-04183-t003:** Distribution of answers between patients and their caregivers.

		Patients Cohort (N; %)	Caregivers Cohort (N; %)	x^2^ *p* Value
**Question 1**				
*Speaking of motor skills, please indicate if in the last year you:*	Remained stable	29; 64.44%	28; 62.22%	(x^2^ (2, N = 90) = 0.461, *p* = 0.794)
	Deteriorated	4; 8.89%	6; 13.33%	
	Improved	12; 26.67%	11; 24.44%	
**Question 2**				
*What do you expect in the next two years*	Stable course	32; 71.11%	22; 48.89%	(x^2^ (2, N = 90) = 4.977, *p* = 0.083)
	Deterioration	2; 4.44%	2; 4.44%	
	Improvement		21; 46.67%	
**Question 3**				
*Would you agree to have participate in a potential clinical trial if, in the absence of side effects or with the possibility of minimal side effects, the prospective was:*	Slow down	29; 64.44%	33; 73.33%	(x^2^ (1, N = 90) = 8.29, *p* = 0.362)
	Stop deterioration	37; 82.22%	36; 80.00%	(x^2^ (1, N = 90) = 0.73, *p* = 0.788)
	Improve	45; 100%	45; 100.00%	N/A
**Questions 4–7**				
Would you consider participating in a clinical trial if it offered the prospect of achieving at least 1, 2, or more than 2 activities on the HFMSE/RULM				
** *HFMSE* ** *Acquisition from score 0* *Not applicable in 3 patients*	I would not participate	3; 7.14%	0; 0.00%	(x^2^ (3, N = 84) = 6.205, *p* = 0.102)
	At least 1 activity	21; 50%	27; 64.29%	
	Two activities	29; 69.05%	30; 71.43%	
	More than 2 activities	39; 92.86%	42; 100.00%	
** *HFMSE* ** *Improvements from score 1* *Not applicable in 6 patients*	I would not participate	5; 12.82%	0; 0.00%	(x^2^ (3, N = 78) = 7.067, *p* = 0.07)
	At least 1 activity	18; 46.15%	22; 56.41%	
	Two activities	24; 61.54%	25; 64.10%	
	More than 2 activities	34; 87.18%	39; 100.00%	
** *RULM* ** *Acquisition from score 0* *Not applicable in 8 patients*	I would not participate	1; 2.7%	0; 0.00%	(x^2^ (3,N = 74) = 4.100, *p* = 0.251)
	At least 1 activity	16; 43.24%	24; 64.86%	
	Two activities	21; 56.76%	27; 72.97%	
	More than 2 activities	36; 97.30%	37; 100.00%	
** *RULM* ** *Improvements from score 1* *Not applicable in 12 patients*	I would not participate	1; 3.03%	0; 0.00%	(x^2^ (3,N = 66) = 1.981, *p* = 0.576)
	At least 1 activity	19; 57.58%	22; 66.67%	
	Two activities	20; 60.61%	24; 72.73%	
	More than 2 activities	32; 96.97%	33; 100.00%	
**Question 8**				
*With the advent of new pharmacological therapies, how has your perception of the disease changed?*	No change	16; 35.56%	13; 28.89%	(x^2^ (2, N = 90) = 0.796, *p* = 0.672)
	More positive	26; 57.78%	30; 66.67%	
	More negative	0; 0.00%	0; 0.00%	
	Uncertain	3; 6.67%	2; 4.44%	
**Question 9**				
*What are your expectations regarding the new pharmacological therapies*	No change	12; 26.67%	6; 13.33%	(x^2^ (2, N = 90) = 3.362, *p* = 0.339)
	More positive	32; 71.11%	37; 82.22%	
	More negative	0; 0.00%	1; 2.22%	
	Uncertain	1; 2.22%	1; 2.22%	

**Table 4 jcm-12-04183-t004:** Population characteristics at the time of the interview subdivided by the year of the questionnaire’s collection.

	Caregiver 2016 (n = 139)	Caregiver 2022 (n = 78)
**Age** ^1^	1.5–17 y (7.38; 3.86)	3 y–17 y (10.08; 3.77)
**Gender** ^2^	63 F (45.32%) 76 M (54.67%)	43 F (55.10%) 35 M (44.87%)
**SMA Type** ^2^		
*SMA Type II*	103 (74.10%)	54 (69.23%)
*SMA Type III*	36 (25.89%)	24 (30.76%)
*Ambulant*	30 (83.33%)	15 (62.50%)
*Non ambulant*	6 (16.66%)	9 (37.50%)

All values refer to the person affected by SMA. ^1^ Range (mean; standard deviation), ^2^ N (%); F = Female, M = Male.

**Table 5 jcm-12-04183-t005:** Analysis of distribution responses to the questionnaire comparing the 2016 sample with the 2022 sample.

		2016 (N; %)	2022 (N; %)	x^2^ *p* Value
**Question 1**				
*Speaking of motor skills, please indicate if in the last year you:*	Remained stable	23; 16.5%	34; 43.6%	(x^2^ (2, N = 217) = 66.88, *p* < 0.001 **)
	Deteriorated	98; 70.5%	10; 12.8%	
	Improved	18; 12.9%	34; 43.6%	
**Question 2**				
*What do you expect in the next two years*	Stable course	30; 21.6%	40; 51.3%	(x^2^ (2, N = 217) = 88.740, *p* < 0.001 **)
	Deterioration	96; 69.1%	3; 3.8%	
	Improvement	13; 9.4%	35; 44.9%	
**Questions 3–6**				
Would you consider participating in a clinical trial if it offered the prospect of achieving at least 1, 2, or more than 2 activities on the HFMSE/RULM				
** *HFMSE* **	I would not participate	4; 2.9%	1; 1.3%	(x^2^ (4, N = 217) = 17.623, *p* < 0.001 **)
*Acquisition from score 0*	At least 1 activity	93; 66.9%	46; 59.0%	
	Two activities	112; 80.6%	48; 61.6%	
	More than 2 activities	129; 92.8%	73; 93.7%	
** *HFMSE* **	I would not participate	8; 5.8%	1; 1.3%	(x^2^ (4, N = 78,217 10.518, *p* = 0.039 **)
*Improvements from score 1*	At least 1 activity	83; 59.7%	44; 56.4%	
	Two activities	104; 74.8%	50; 64.10%	
	More than 2 activities	119; 89.6%	68; 87.2%	
** *RULM* **	I would not participate	3; 2.2%	1; 1.3%	(x^2^ (4, N = 217) = 12.156, *p* = 0.016 **)
*Acquisition from score 0*	At least 1 activity	69; 49.6%	36; 46.2%	
	Two activities	82; 59.00%	40; 51.3%	
	More than 2 activities	92; 66.2%	58; 74.4%	
** *RULM* **	I would not participate	15; 10.8%	1; 1.3%	(x^2^ (4, N = 217) = 30.160, *p* < 0.001 **)
*Improvements from score 1*	At least 1 activity	61; 43.9%	46; 59.0%	
	Two activities	80; 57.6%	49; 62.8%	
	More than 2 activities	89; 64.1%	68; 87.2%	
**Question 7**				
*Would you agree to have your child participate in a potential clinical trial if, in the absence of side effects or with the possibility of minimal side effects, the prospect was to slow down or to improve or to stop deterioration of motor function:*	Slow down			
	Yes	99; 71.2%	55; 70.5%	(x^2^ (2, N = 217) = 4.330, *p* = 0.115)
	No	17; 12.2%	16; 20.5%	
	Uncertain	23; 16.5%	7; 9.00%	
	Stop deterioration			
	Yes	121; 87.1%	62; 79.5%	(x^2^ (2, N = 217) = 3.272, *p* = 0.195)
	No	7; 5.0%	9; 11.5%	
	Uncertain	11; 7.9%	7; 9.00%	
	Improvement			
	Yes	134; 96.4%	75; 96.2%	(x^2^ (2; N = 217) = 0.552; *p* = 0.759)
	No	2; 1.4%	2; 2.6%	
	Uncertain	3; 2.2%	1; 1.3%	

** = significant difference.

## Data Availability

Individual data are stored in each centre and a cumulative database with individual data is stored by the corresponding author. The datasets generated and/or analysed during the current study are not publicly available due to the fact that consent was obtained to provide only cumulative anonymized data but are available from the corresponding author on reasonable request.

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
