# Peer review of "Caregivers’ Expectations on Possible Functional Changes following Disease-Modifying Treatment in Type II and III Spinal Muscular Atrophy: A Comparative Study"

_jcm, 2023, doi:10.3390/jcm12134183_

Round 1

Reviewer 1 Report (New Reviewer)

Overall: There are multiple errors with grammar that need to be corrected.  Many are not listed below (ex: line 297: "In an historical..." should be "In a historical..."

Page 1 Line 3=29-30: In the conclusion, please give a conclusion beyond the expected secondary use of the data going forward.

Introduction: Why is there no discussion of type 1 SMA?

Methods: As you only survey patient's with type II and III SMA, this should be included in the abstract and title.

Line 194-195: Were any of the caregivers the same between the two studies?  How was this methodology accounted for?  Was there direct comparison of answers between the two timeframes?

There was no comparison of patient's perceptions between 2016 and 2022, thus the title is not correct (it includes patient and caregiver).

Page 2 line 78-79: This is not true that age 13 is when patients are asked to assent as it can be as low as age 7 in the US.  As such, please specify given this statement.

For the percentage of professional, non-parent, caregivers, please specify their educational and professional backgrounds as that may affect their responses.  Similarly, are any of the parents and/or patients in the healthcare field or associated scientific fields for professions?  Their baseline educational backgrounds are important for this type of study as the sample size may be significantly affected.

Page 5: The statement that there were no significant differences between patient and caregivers perspectives is not clear. Was this done as a direct comparison (a patient was compared to his or her caregiver) or cohort comparison (entire groups compared)?

Line 194: requires a "-" in Seventy-eight

Line 216-218: "The positive outcome of the available therapeutic approaches for type II and III SMA pa-216 tients in clinical trials [20-22] and real world data [23-25] has highlighted the need to es-217 tablish if the changes observed on the functional scales are associated with carers’ and 218 patients’ perspective [26-31]."  

This statement implies that the perspective causes the changes of functional scales which is not within the scope of this article.  The statement would be better written:
"The positive outcome of the available therapeutic approaches for type II and III SMA pa-216 tients in clinical trials [20-22] and real world data [23-25] hightlights the need to understand if there is a change to caregiver's perceptions of the disease."

Line 223-224: The paper does not compare 2016 to 2022 patient characteristics but only caregiver characteristics.

Regarding the population of patients, were all patients on a disease-modifying therapy?  If not, how can you account for the variations outside of the variation in potential patients being treated.  Additionally, how long were patients on disease modifying therapies and what therapies they were on should be included in this study.   Additionally, it was not clear from the paper that in the first cohort of patients, no patient was in a trial for disease modifying therapy.

Clear inclusion and exclusion criteria should be written in the methodology section.

Discussion:
Multiple times, the methodology is reiterated which is not appropriate for this section.  Please rewrite to have a clear discussion.

Line 265-266: Please clarify if the lack of variance between patient and caregiver was as written or if it was a lack of variance between the cohorts.

Line 266: "At variance..." is not proper grammar, please correct.

Line 284: This is incorrect as there was no patient's studied (per author's) in 2016 so cannot conclude there was improvement in perceptions.

Line 289-292: This is outside the scope of the study as a conclusion, please remove.  Perceptions of the scale were not evaluated as a viable caregiver opinion on an individual basis but relative to patients.  To make this conclusion, direct assessment of individual items would be need to had not a point scale gains assessment as value only as that would vary amongst the items.

There are various grammatical and word-choice errors that need to be corrected throughout the paper.  

Author Response

Reviewer 2 Report (New Reviewer)

The study by Pera & al. presents a study on the perceptions of caregivers and SMA patients concerning DMTs in SMA. This is, in some way, a follow-up study of their initial study that was performed prior to DMTs availability and serves as a great longitudinal tool to see how attitudes may be changing the SMA patient community over time. The study is simple, of interest. I provided a few suggestions/concerns, below that would strengthen/improve the clarity of the manuscript. More details are presented below:

Major comments

Study design

·      How was N number arrived at? Was there any sample size calculation performed?

·      How was the missing data dealt with? (if any?).

·      Consideration to stratify patients by time from diagnosis would be important. The longer disease is established, the fewer MN present, and expectations would likely be lower (at least for clinicians). It would be of interest to know if this is also the case from a caregiver/patient perspective. Indeed, if the results showed that patients with long-established diseases were expecting significant improvement, it would raise the alarm that, as a medical community, we should do better in educating our patients about expected outcomes in different scenarios (this should also probably be discussed in the discussion of the manuscript).

·      I would be interested to know how professional caregivers are defined. This makes me raise questions about how they could answer the survey questions concerning clinical trial participation for a patient with whom they have only professional ties and would have no power in the decision-making of the patient for enrollment or inclusion in a study.

Population

·      This is for the most part an adult population (patient in Table 1). Consider describing (in the demographic data table) each subset of adult and pediatric for both patients and caregivers in the characteristic table, especially as results are later used to compare both groups.

·      It would be important to describe whether patients are receiving DMTs (were they all receiving DMTs?) in the baseline characteristic as this could lead to some amount of bias toward their views about treatment (and past experiences). A comparison between those two groups (receiving DMTs VS not receiving DMTs) would be helpful.

Results

·      While it did not reach significance, Q2 in Table 3 does show that patients may be more cautious compared to their optimistic caregivers. I believe this might be worth noting.

·      It might be worth doing a sensitivity analysis/separate analysis for “Comparison between 2016 and 2022 caregivers responses” with only the 78 in the study in 2016. It would provide a much more authentic view of the pre and post expectations. Some of the caregivers in the new cohort may only have known a time when SMA had DMTs available with prior knowledge of the effect of such DMTs in clinical trials.

·      It might be worth discussing that patients informally seem to have positive views towards DMTs even if it may not appropriately be captured by HFMSE or RULM such as less fatigue, better/easier ADLs, or others. The gross motor scale may not be the only outcome that the patient would look upon to perceive therapy.

Minor comments

Line 24 – Rephrase sentence – Currently does not make sense and the word/variation of include twice in the same sentence.

Line 31 – “differences in 6 of 7 questions” is not very informative. It should probably be changed by the goal of the question/findings (ie. Perception, etc.)

Line – 58-59 – please rephrase – there is currently “use the same questionnaire” twice in the same sentence

Line 64-66 – “The two centres shared the same training and had already performed inter-observer reliability for the Hammersmith Motor Functional Scale Expanded (HFMSE) and Revised Upper Limb Module scales (RULM) as part of clinical trial studies [14].” Does this statement matter? These scales were not used to compare patients between sites in the study, but rater survey based? I would consider removing it.

Line 194 – Seventyeight – Consider putting in numbers or correcting spelling.

Lines 265-268 – “we also administered the questionnaire to patients older than 13 years and, 267 independently, to their carers.” This is a repetition of what is written a line or two above.

Line 268 – “At variance” is used twice in two subsequent sentences.

See above in the Comments and suggestions for authors.

Author Response

Reviewer 3 Report (New Reviewer)

Thank you for this very interesting and important study reporting patients' and caregivers' perspective of the changing landscape and future of SMA.

Comments and suggestions:

General:

1. Did you collect any information regarding treated vs. non-treated, and type and timing of treatment for patients in your current study? This is important information and should be presented along with the other population characteristic data.

Introduction:

1. In the introduction, you mention SMA phenotypes. I think it is important to emphasize that the historical classifications for SMA phenotypes are changing with the introduction of disease modifying treatments, and that patients previously classified as Type I or Type II may achieve skills (such as walking) that previously defined individuals as Type III, so that the lines between these categories are now significantly blurred.

Methods: 

1. Along the same lines as above - please clarify how you defined SMA Type II and Type III for the purposes of the current (post-DMT) study. 

2. Please clarify how you defined "ambulatory" for the purposes of this study.

Results:

1. Tables 1 and 4: Please subcategorize ambulant vs. non-ambulant patients according to SMA type. From the numbers in these tables, it appears that there were no ambulatory SMA Type II patients in your study, but this is unclear from the layout of the tables. Please indent the ambulant/non-ambulant categories or otherwise make this easier to understand.

Just a few very minor edits to grammar needed (for example, page 6, line 194 "Seventyeight" needs a space). I feel it just needs one more close review!

Author Response

This manuscript is a resubmission of an earlier submission. The following is a list of the peer review reports and author responses from that submission.

Round 1

Reviewer 1 Report

The present cohort study aimed to explore the pre - and post-treatment changes in SMA patients' and caregivers' expectations on possible functional changes following treatment. A structured questionnaire was used to explore their perspective and expectations in relation to current or future clinical trials. This is interesting and has implications for future experimental design and interpretation of real-world data. But there are still some unclear or mistake points. The comments are as follows:

Abstract section,

1.       A brief description of the composition for “A questionnaire” is required.

Materials and Methods

2.       Chi-square test/Fisher’s test is not suitable for all data.

Results section,

1.      The form of the table needs further check.

2.      Significant difference exists only where indicated by **, other data needs to be added to represent the meaning and discussed in the context of the literature.

3.      What kind of people are included in the caregivers mentioned in the article, does it only refer to parents? Is there a difference between the parents mentioned in the article? (line78, line110, line 236)

Discussion section,

1.      The discussion section lacks necessary literatures(line212-240). Please discuss the reasons for the results of this study in the context of the literature.

2.      References 19-24 are repeatedly cited in the Discussion section, and these do not support all of the discussion.

3.      The conclusion needs to be more concise and reflect the main idea of this study. Limitations do not fit in the conclusion section.

Reviewer 2 Report

This study by Pera et al. focusses on the perception of SMA patients (as from age 13) and their caregivers concerning motor skills, clinical changes and their attitude to future clinical trials in the era of now availible new therapies. Results are compared to a similar study conducted by an overlapping study group in 2016 where also caregivers of SMA patients were interviewed. The former study focused on the patients and caregivers views on the clinical relevance of the Hammersmith Functional Motor Scale Expanded- (HFMSE). The former study was published in BMC Neurology (IF 2.17).

As expected, SMA patients in 2022 reported better motor performance under the availible therapies and had positive expectations for the future trend of the disease. It does not astonish the reader that the idea of taking part in a clinical trial is an option for expected larger steps in motor improvement since stability is already achieved for many patients.

The study results are well presented. I would recommend to explain in more details if any of the interviewed persons in 2022 were already part of the 2016 study. It is also unclear why the age of 13 was chosen for patients self reports.

I can not see a clear novelty or achievement in the presented data, especially not for publication in a journal with double the impact factor of the former publication, which presented several noveltys in the evaluation of a questionnaire and the clinical meaningfulness of the questionnaire in self reported evaluation.

The actual repetition of the study  adds minimal novelty in a field of emerging therapies and is presenting data that looks like commonplaces.

Round 2

Reviewer 1 Report

The authors responded to my previous comments, but did not address my query. Unfortunately, I don't see the author's revision of this manuscript, only the response to the previous comment. In the response, the authors did not explain well the inadequacy of the design of the methods and results section. The authors did not explain whether all data qualified for the chi-square test. Many results without variability take up too much length, and the results section needs to be further condensed to focus on the meaningful results. As well, the discussion section needs to address the innovation and significance of this study by analyzing what the results represent in the context of the larger literature, which is not evident in the current manuscript.

Since the changes to the manuscript made by the authors could not be seen, there is no sufficient reason to agree it for publication.

Please provide the revised manuscript and explain in detail what amendments have been made.

I have carefully read the authors' responses and the revised version of the manuscript, but the authors did not go into depth to revise the shortcomings of the article. I hope the authors understand that the comments I gave are for the improvement of the manuscript and better presentation of the results.

The author only changed a few sentences that I mentioned previously, so the sincerity of the author in revising the manuscript was not seen, or perhaps the author stubbornly stuck to their current writing.

The manuscript is just a repetition of a study already done in 2016 (same questionnaire, 58 of the 150 caregivers had previously participated in the 2016 study), without enough innovations and findings. This is also indicated by the large number of meaningless results that can be seen in the results section, with few results with significant differences (only where were indicated by **).

Also, the discussion section is poorly written, with one literature (13) or multiple literatures (20-25) being repeatedly cited in different places. The authors seem unwilling to discuss the results found in this study in more depth to show what these findings represent for the research progress in SMA.

The conclusion section, still describes the meanings of the results, which should probably be shown in the discussion, thus a final and clear conclusion about this study cannot be drawn.

Author Response

RESPONSE TO REVIEWER 1 COMMENTS

Comment

“The authors responded to my previous comments, but did not address my query. Unfortunately, I don't see the author's revision of this manuscript, only the response to the previous comment. In the response, the authors did not explain well the inadequacy of the design of the methods and results section. The authors did not explain whether all data qualified for the chi-square test. Many results without variability take up too much length, and the results section needs to be further condensed to focus on the meaningful results. As well, the discussion section needs to address the innovation and significance of this study by analyzing what the results represent in the context of the larger literature, which is not evident in the current manuscript. Since the changes to the manuscript made by the authors could not be seen, there is no sufficient reason to agree it for publication. Please provide the revised manuscript and explain in detail what amendments have been made.”

We understand that the reviewer 1 had not received the latest version of the revised manuscript, therefore he could not check if the amendments to his requests were done.

In the response, the authors did not explain well the inadequacy of the design of the methods and results section. The authors did not explain whether all data qualified for the chi-square test. Many results without variability take up too much length, and the results section needs to be further condensed to focus on the meaningful results.

These information’s were all included in the revised manuscript that the reviewer has not received. In particular:

Chi square test: line 105-110 “Chi-square test/Fisher’s exact test of independence was used to compare the questions responses distribution between age (paediatric vs adult), participants status (caregivers, patients), motor functional status (ambulant, non ambulant), pre-post pharmacological treatment era (2016, 2022). As per statistical procedure, we used the Chi-square test if all of the expected cells were greater than 5, while we used the Fisher's exact test of independence if an expected number was less than 5.”

Design and methods: line 174-179 “ The choice of 13 years is often considered as the age when teenagers have ability to pro-vide meaningful answers. Many psychometric studies also use the age of 13 to collect data on minor patients as from this age there is more awareness of the disease, making the re-sponses more reliable than those obtained in younger children [19]. Furthermore, in clini-cal trials, the age of 13 is the one when minors are asked to sign the study.”

And other corrections were done through the results section and tables have been amended.

“As well, the discussion section needs to address the innovation and significance of this study by analyzing what the results represent in the context of the larger literature, which is not evident in the current manuscript. Since the changes to the manuscript made by the authors could not be seen, there is no sufficient reason to agree it for publication. Please provide the revised manuscript and explain in detail what amendments have been made.”

The innovation and the significance of this study had been explained in the response to reviewer 2 and included in the version of the manuscript that reviewer 1 has not received. In particular, in the introduction we underlined that we are in an historical moment of strong change for SMA and it is essential to highlight the point of view of patients and caregivers with respect to the expectations towards the therapies currently used and to the meaningfulness of functional abilities in their daily life. The need to capture patients and caregivers perspective on the impact of the new treatments is not only acknowledged by clinicians and families but has also been strongly encouraged by the United States Food and Drug Administration (FDA), by the European Medicine Agency (EMA) and endorsed by the SMA community. 

To add to this point, we also want to acknowledge with the reviewer that the novelty of our research article is that we have shown for the first time the changes over time: the aim of the paper was to compare the pre and post treatment findings. We are not aware of any other longitudinal study reporting patient perspective in the pre-treatment era and after a few years of treatment and while a possible change could have been foreseen, it has never been systematically explored. The evidence that this finding is of interest for the SMA community  is proven by the fact that the precedent paper collected  nearly 80 citations in just a few years. We think our findings will provide a better understanding of the patients’ view on this topic and facilitate discussion with regulators.

I have carefully read the authors' responses and the revised version of the manuscript, but the authors did not go into depth to revise the shortcomings of the article. I hope the authors understand that the comments I gave are for the improvement of the manuscript and better presentation of the results.

The author only changed a few sentences that I mentioned previously, so the sincerity of the author in revising the manuscript was not seen, or perhaps the author stubbornly stuck to their current writing.

The manuscript is just a repetition of a study already done in 2016 (same questionnaire, 58 of the 150 caregivers had previously participated in the 2016 study), without enough innovations and findings. This is also indicated by the large number of meaningless results that can be seen in the results section, with few results with significant differences (only where were indicated by **).

Also, the discussion section is poorly written, with one literature (13) or multiple literatures (20-25) being repeatedly cited in different places. The authors seem unwilling to discuss the results found in this study in more depth to show what these findings represent for the research progress in SMA.

The conclusion section, still describes the meanings of the results, which should probably be shown in the discussion, thus a final and clear conclusion about this study cannot be drawn.

We completely agree with the reviewer comments. We acknowledge that our paper, as it was presented, was out of focus and not showing clearly all the results of the study. To address the reviewer comments, we have slightly modified the title of the paper, underling at first sight the aim of the paper. We want to make clear to the reader that the primary aim of this study was to report the results from the comparative analysis performed between the 2016 and 2022 studies. Other than that, we have also re-shaped the methods and results section. In the methods now is clearly stated that the study was composed by 2 parts. One introductory, aiming to solely describe the 2022 responses, and one dedicated to the primary aim analysis. In the results section now there are also 2 subtitles dividing them in the two parts, so the reader can understand the purpose of the performed analysis. We feel now that this subdivision successfully highlight the positive results of our primary analysis, with 6 out of 7 questions showing differences between the two era (2016 vs 2022).

Furthermore, following this changes, the discussion is now focused on both part, showing to the reader also the non-significant results and explaining why “non-statistically significant” doesn’t mean that the results are not meaningful. As suggested by the reviewe, we have added several references in the discussion section to go deeper into the meaning of our results and demonstrate the innovation and findings of the study. As suggested by the reviewer, we have also changed the conclusion, focusing more on the innovation that this study provides to the community.